

**Assessing the extreme risk of coastal inundation due to climate**
**change: A case study of Rongcheng, China**
Aiqing Feng[1,2], Jiangbo Gao[1], Shaohong Wu[1*], Yanzhong Li[2,3], Xiliu Yue[1,2]
[1] Key Laboratory of Land Surface Pattern and Simulation, Institute of Geographical Sciences and Natural Resources
Research, Chinese Academy of Sciences, Beijing 100101, China;
[2] University of Chinese Academy of Sciences, Beijing 100049, China;
[3] Key Laboratory of Water Cycle and Related Land Surface Processes, Institute of Geographical Sciences and Natural
Resources Research, Chinese Academy of Sciences, Beijing 100101, China.
*Corresponding author: Shaohong Wu
E-mail: wush@igsnrr.ac.cn



**Abstract**: Extreme water levels, caused by the joint occurrence of storm surges and high tides,
always lead to super floods along coastlines. Given the ongoing climate change, this study explored
the risk of future sea-level rise on the extreme inundation by combining P-III model and losses
assessment model. Taking Rongcheng as a case study, the integrated risk of extreme water levels
was assessed for 2050 and 2100 under three Representative Concentration Pathways (RCP)
scenarios of 2.6, 4.5, and 8.5. Results indicated that the increase in total direct losses would reach
an average of 60% in 2100 as a 0.82 m sea-level rise under RCP 8.5. In addition, affected population
would be increased by 4.95% to 13.87% and GDP (Gross Domestic Product) would be increased by
3.66% to 10.95% in 2050 while the augment of affected population and GDP in 2100 would be as
twice as in 2050. Residential land and farmland would be under greater flooding risk in terms of the
higher exposure and losses than other land-use types. Moreover, this study indicated that sea-level
rise shortened the recurrence period of extreme water levels significantly and extreme events would
become common. Consequently, the increase in frequency and possible losses of extreme flood
events suggested that sea-level rise was very likely to exacerbate the extreme risk of coastal zone in
future.
**Keywords:** sea-level rise; inundation risk; extreme water level; expected direct losses; affected
population and GDP; recurrence period.

## 1 Introduction

Coastal inundation is predominantly caused by extreme water levels when storm surges are
concurrent with astronomical high tides (e.g. Pugh, 2004; Quinn et al., 2014). Statistically, the
extreme flood events were occurred frequently and caused huge devastation (Trenberth et al., 2015).
Recent research indicated that sea-level rise, with global mean rates of 1.6 to 1.9 mm yr$^{-1}$ over the
past 100 years (Holgate, 2007; Church and White, 2011; Ray and Douglas, 2011), had been strongly
driving the floods (Winsemius et al., 2016). Global mean sea-level was expected to rise more than
1 m by the end of this century (Levermann et al., 2013; Dutton et al., 2015), even if global warming
can be controlled within 2℃. Thus, coupled with continuous sea-level rise induced to climate
change, the future coastal inundation risk in terms of hazards and possible losses should be paid



attention to disaster mitigation.
Projections for extreme water levels are indispensable for inundation risk assessment. Most
researches to date have focused on the coastal flooding caused by storm surges (e.g. Bhuiyan and
Dutta, 2011; Klerk et al., 2015). At present, exceedance probabilities of current extreme water level,
induced by tropical and extra-tropical storm surges, have been estimated (Haigh et al., 2014a, b).
However, on account of the sea-level rise, coastal flooding disasters would become more serious
(Feng et al., 2016) and 85% of global deltas experienced severe flooding in recent decades (Syvitski
et al., 2009). Feng and Tsimplis (2014) showed that extreme water level around the Chinese
coastline was increased by 2.0 mm to 14.1 mm yr$^{-1}$ from 1954 to 2012. Based on an ensemble of
projection to global inundation risk, it argued that the frequency of flooding in Southeast Asia is
likely to increase substantially (Hirabayashi et al., 2013). By 2030, the portion of global urban land
exposed to the high-frequency flooding would be increased to 40% from a 30% level in 2000
(Guneralp et al., 2015). Conservative projections suggested that over a half of global delta surface
areas would be inundated as a result of sea-level rise by 2100 (Syvitski et al., 2009).
The impacts of coastal flooding on social economies were considered and some methods were
established to estimate the possible losses (e.g. Yang et al., 2016). With the socio-economic
development, the large aggregations of coastal population and assets would lead to the increase
exposed to inundation in future (Mokrech et al., 2012; Strauss et al., 2012; Alfieri et al., 2015).
Without adaptation, by 2100, 0.2% to 4.6% of the global population would be at risk of flooding,
and expected annual GDP losses would be 0.3% to 9.3% (Hinkel et al., 2014). In particular,
urbanization of China was rapidly fast in the world and many low-lying coastal cities were
confronted with high probabilities of flooding (Nicholls and Cazenave, 2010). More than 30% of
the China's coast was assessed as 'high vulnerability' according the research of Yin et al., (2012),
and the population numbers exposed to flooding risk were the highest in the world (Neumann et al.,
2015). A number of China's cities including Guangzhou, Shenzhen, and Tianjin were in the top 20
global cities in terms of their exposure to 100-year inundation risk and huge average annual losses
because of water levels rising (Hallegatte et al., 2013).
Distinguishing the risk of extreme floods considering sea-level rise caused by climate change
is vital for disaster mitigation and adaptation on a large time scale. In this study, the flooding from
extreme water levels was simulated by a combination of storm surges, astronomical high tides, and



sea-level rise heights under different RCP scenarios. Using Rongcheng City as a case study, a
comprehensive multi-dimensional analysis was presented to assess the inundation risk based on two
time scales of 2050 and 2100, and three RCP scenarios of 2.6, 4.5, and 8.5. The main objectives are
to (1) investigate the expansion of the inundated area and the increase in expected direct losses; (2)
analyze the effect of sea-level rise on population and GDP; and (3) reveal the future hazard change
of extreme water levels by the probability of occurrence.

## 2 Data and methodology

### 2.1 Study area

Rongcheng City, located at the tip of the Shandong Peninsula, China, is surrounded on three sides
by 500 km of Yellow Sea coastline (Fig. 1). This city has low-elevation and flat topography and
covers an area of more than 1,500 km$^2$. Its population of 0.67 million people and GDP of \$12.31
billion make it become one of the top one hundred counties in China. Rongcheng experiences a
monsoonal climate at medium latitudes with an average annual rainfall of 757 mm and a temperature
of 11.7℃ for nearly 50 years (data from http://data.cma.cn/). It is also in a critical geographical
position for trade exchange and the modern economy facing Korea across the Yellow Sea.
Substantial additional capital investment is expected in this region because the Shandong Peninsula
National High-tech Zone has been approved as a part of the National Independent Innovation
Demonstration Zone by the China's State Council in 2016 (http://www.gov.cn/). A inundation risk
assessment for Rongcheng City is urgent to its long-term development, especially under the situation
of sea-level uptrend due to climate change.




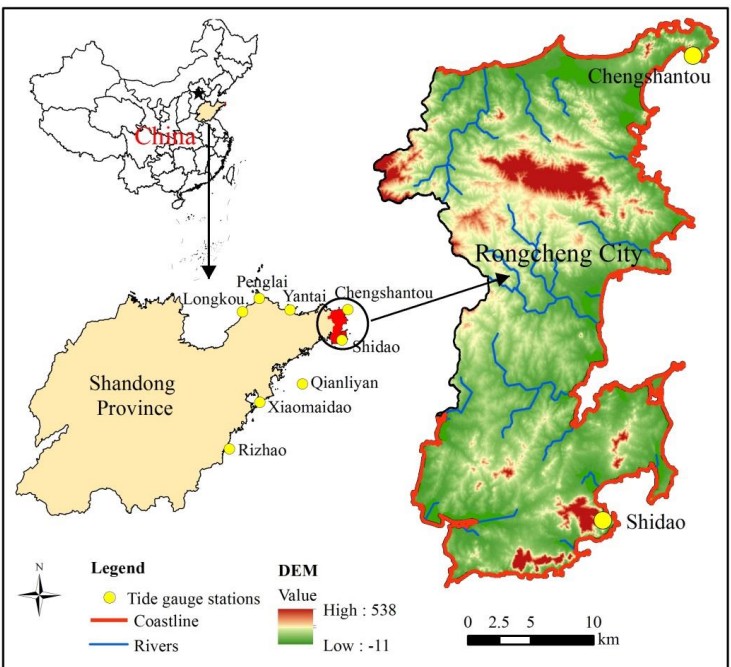


**Fig. 1** Map to show the geographic locations of Rongcheng City and main tidal gauge stations


**2.2 Assessment process and dataset**
The assessment process of inundation risk followed three steps. First, extreme water levels were
calculated using storm surge data, astronomical high tides, and sea-level rise heights by the method
of Pearson Type Ⅲ (P-Ⅲ). Second, the inundated area and depth were identified by the flood
model (the four nearest neighbors algorithm) using the data of extreme water levels which resulted
from the first step and the Digital Elevation Model (DEM). Third, inundation risk was assessed by
direct losses model and recurrence period change. The dataset was summarized in Table 1.










**Table 1** Dataset of extreme risk assessment including hydrological, geographical, and statistical inputs

| Data type | Content | Description | Source |
|---|---|---|---|
| Hydrological data | Sea-level rise | Global mean sea-level rise in 2050 and 2100 under RCPs 2.6, 4.5, 6.0, and 8.5. All scaled with two degrees (low vs. high) | IPCC (2013) |
| | Storm surge | Return periods of storm surges were obtained using the P-III model and historical data from 1967 to 2013 | Tidal gauge stations, National State Oceanic Administration |
| | Astronomical high tide | Predicted using harmonic tide models based on measured data (Wu et al. 2016) | Tidal gauge stations, National State Oceanic Administration |
| Geographical data | 1:10,000 digital topographic maps | A 10 m × 10 m DEM was built using elevation points and contour lines in ArcGIS | Bureau of Land Management |
| | Land-use maps | High precision grid data at a 30 m scale were used to characterize the land-use types in flooded area and calculate direct damage | Institute of Geographical Sciences and Natural Resources Research, Chinese Academy of Sciences (IGSNRR, CAS) |
| | Spatial distribution of GDP and population | 1 km × 1 km grid data according to statistics from 2010 | http://www.resdc.cn/ |
| Statistical data | Vulnerability curves and estimated loss values for different land-use types. Abbreviations: y, loss rate (%); x, flood depth (m); V, loss values ($/m²). | Residential land, y=16.682x, $R^2$=0.6359, V=307.69; Farmland, y=49.837x, $R^2$=0.4246, V=0.77; Grassland and woodland, y=36.304x, $R^2$=0.9113, V=12.31; Unused land and water regions, y=0, V=0. | (Yin, 2011) |


### 2.3 Construction of the cumulative probability distribution of extreme water levels

Extreme water level is a compound event caused by storm surges and astronomical high tides while
sea-level rise also contributes to extreme water levels under global climate change. Therefore, in
this study, the current extreme water levels (CEWLs) and future extreme water levels were
constructed. The latter was a combination of CEWLs and projected heights of sea-level rise under
different RCP scenarios and was defined as the scenario extreme water levels (SEWLs). The
cumulative probability distribution curves of CEWLs and SEWLs were refitted using a P-Ⅲ model
as the Equation (1). The details of this method were shown as Wu et al., (2016).


$$f(x) = \frac{\beta^\alpha}{\Gamma(\alpha)} \int_{x_p}^{\infty} (x - \alpha_0)^{\alpha-1} \ e^{-\beta(x-\alpha_0)}$$
(1)

In this expression, $\alpha$, $\beta$, and $\alpha_0$ are the shape, scale, and location parameters, respectively; $x$ is
the annual maximum values for water levels; $p$ is the probability of occurrence.
$$CEWL = ST + AHT$$
(2)

where $ST$ is storm surge and $AHT$ is astronomical high tide;
$$SEWL = CEWL + \ SLR$$
(3)

where $SLR$ is the predicted height of sea-level rise in the future;
$$T = 1/p$$
(4)

where $T$ stands for the recurrence period of extreme water level and the $T$-year recurrence level
means that an event of extreme water level has a $1/T$ probability of occurrence in any given year
(Cooley et al., 2007).
Because of the uncertain impacts of sea-level-rise on storm surges, the statistical probabilities
of storm surge in this model were assumed to be unchanged in future (e.g. Hunter, 2012; Kopp et
al., 2013; Little et al., 2015). The extreme water levels were mainly constructed by historical records
of Chengshantou and Shidao tidal stations located in Rongcheng City (Fig. S1 in Supplementary
data). In order to reduce the error caused by the spatial distribution of extreme water levels, recorded
data of the surrounding six tidal stations (including Longkou, Penglai, Yantai, Qianliyan,
Xiaomaidao, and Rizhao) on Shandong Peninsula were still calculated using the inverse-distance-
weighted (IDW) technique in ArcGIS software.
**2.4 Identification of flooding**
Inundated area was extracted from the flood model using the four nearest neighbors algorithm based
on high-resolution DEM (10 m × 10 m) and extreme water level layers (10 m × 10 m cells generated
in ArcGIS). Flooding criteria were that the extreme water level of layer cells must be greater than
or equal to the elevation of DEM and inundated cells must be connected to the coast individually
(Xu et al., 2016). The impacts of the elevations of urban landscapes and other buildings on flooding
process were not considered in this study. In this section, inundated area and depth could be
computed.



### 2.5 Inundation risk assessment

Expected direct losses were calculated using inundated area, inundated depth, vulnerability curves, and loss values for each land-use type. The land-use map of 30 m resolution was resampled to 10 m cells using the raster processing tool in ArcGIS in order to match inundated cells. The assessment model for expected direct losses is:

$$EDL = \sum_i A \times h \times r \times V \tag{5}$$

where $EDL$ stands for the expected direct losses of extreme floods; $i$ denotes land-use type including residential land, farmland, woodland, grassland, and unused land; $A$ denotes inundated area; $h$ stands for flood depth; $r$ stands for loss rate (vulnerability curves); and $V$ stands for the per-unit loss value ($/m$^2$).

The amounts of affected population and GDP were estimated based on the grid distribution data of population and GDP (published in China 2010 at a resolution of 1 km, http://www.resdc.cn/). Land-use cover change and socio-economic development were not considered in future (Hallegatte et al., 2013; Hinkel et al., 2014; Muis et al., 2015).

## 3 Results and analysis

### 3.1 Inundated area

In the absence of adaptation, the areas inundated by CEWLs and SEWLs are shown as Fig. 2. At the present stage, inundated areas range from 156.60 km$^2$ to 168.8 km$^2$ when Rongcheng City encounters extreme water levels. However, an expanding trend in inundated area is inevitable because of future sea-level rise; in this analysis, the smallest increase in inundated area would be seen under RCP 2.6 while the largest would be seen under RCP 8.5 while it would be enlarged significantly by 2100 compared to 2050 as sea-level rise continues. The extreme scenario, under RCP 8.5, predicts that the total area where were threatened by flooding ranges from 168.35 km$^2$ to 186.46 km$^2$ in 2050, and that it may be between 187.72 km$^2$ and 199.18 km$^2$ by 2100. According to this projection, the maximum area is around 13% by the end of the century. At high degree for each RCP scenario, inundated area increases by 2100 is likely to range from 14.21% to 19.54% given a 100-year recurrence. Summary statistics of future inundated area increase for 50 to 1,000-year recurrence periods are presented in Table S1(a).


172

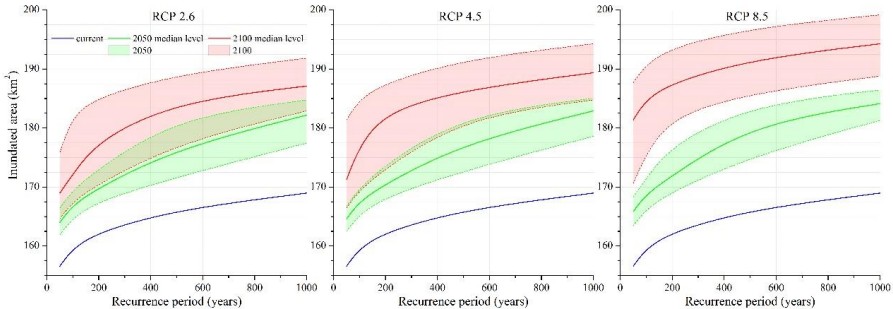

173

**Fig. 2** Inundated areas under different RCP scenarios for 2050 and 2100. The blue solid line denotes the inundated

area curve as it changes with CEWLs, while the areas outlined by green and red stippled lines denote the extent of

inundated areas projected on the basis of SEWLs under low and high degree RCP scenarios for 2050 and 2100,

respectively. The green and red solid lines denote the median degree for each RCP scenario. Similarly, the

explanations are used for Fig. 4 and 5.

Land-use types of residential land, farmland, woodland and grassland are involved in the

estimation of total inundated area while the water bodies and unused land could be ignored in this

study. Thus, summarizing the inundated data, the total inundated land-use areas under RCP 8.5 are

shown in Fig. 3. Results show that residential land and farmland are more exposed to extreme water

levels than woodland and grassland. Indeed, when Rongcheng City is currently subjected by

extreme flooding, 42.63 km$^2$ to 46.77 km$^2$ of residential land and 34.15 km$^2$ to 39.97 km$^2$ of farmland

would be affected, based on 50 to 1000-year recurrence periods, respectively. Given a high degree

RCP 8.5 scenario, inundated areas of residential land and farmland would increase to 47.61 km$^2$ and

41.13 km$^2$ in 2050, and to 52.88 km$^2$ and 51.47 km$^2$ in 2100, respectively. More seriously, combined

areas of residential land and farmland exposed to flooding would rise to around 50 km$^2$ in 2050 and

56 km$^2$ in 2100, respectively. The flood map (Fig. S2) shows the extension of inundated area by

2050 and 2100 given a 100-year recurrence period.



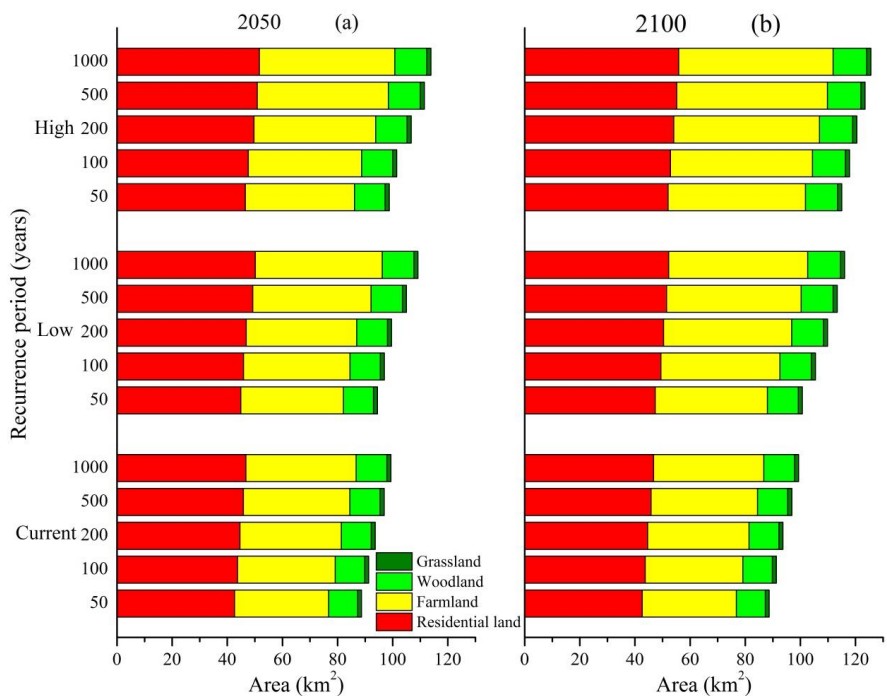

**Fig. 3** Predicted inundated areas broken down by different land-use types given 50 to 1,000-year recurrence
periods in 2050 (a) and 2100 (b). RCP 8.5 is taken as an example in this paper and the inundated areas of different
land-use types under RCP 2.6 and 4.5 are similar.

**3.2 Expected direct flood losses**
Flood damage does not only depend on inundated area and depth, but is related to the loss rates and
values of exposed land-use types. The total expected direct flood losses would be exacerbated with
sea-level rise (Fig. 4), but for current extreme floods, loss magnitudes are up to $0.53 billion and
$0.69 billion for 50 to 1,000-year recurrence period CEWLs. Predictions for future extreme flood
show an increase of more than 20% when the elevation of sea-level rise exceeds 0.3 m, however,
the increase rates expand to beyond 40% given a 0.5 m sea-level rise. Indeed, by 2050, estimated
losses under the RCP 2.6 scenario would be between $0.6 billion and $0.84 billion. These losses
would be slightly increased by 2050 under the RCP 4.5 and 8.5 scenarios. Analyses show that
expected direct losses would be more aggravated by the end of the century. By 2100, the smallest


range of expected damage given the low degree RCP 2.6 scenario would be between $0.63 billion
and $0.81 billion. However, the maximum range of expected damage under the high degree RCP
8.5 scenario is predicted to be between $0.88 billion and $1.08 billion. It is worth noting that the
increase rates reach an average of 60% under the high degree of RCP 8.5 scenario with a 0.82 m
sea-level rise. The largest increase in predicted direct flood damage would be up to 29% in 2050
and 67% in 2100. Additive statistical information of future expected direct losses increase is
presented in Table S1(b). The losses for main land-use types under the high degree RCP 8.5 scenario
are shown in Table S2 and results indicated that residential land would be seriously affected by
extreme floods.

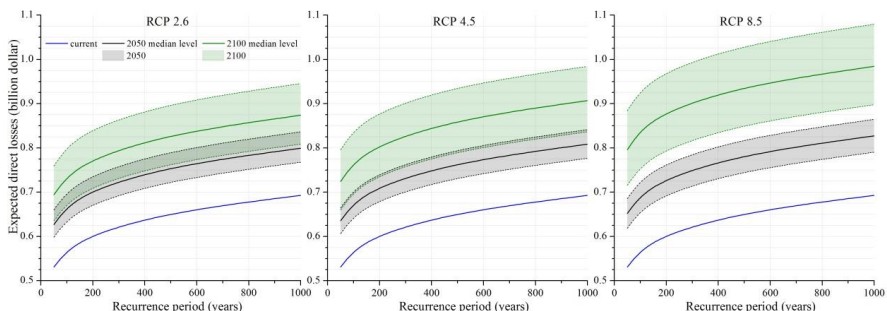


**Fig. 4** Expected direct losses (billion, dollar) in 2050 and 2100 given different RCP scenarios.


**3.3 Population and GDP affected by extreme water levels**
With the rapid socio-economic development, population and GDP have distributed along the
coastline. Thus, a large proportion of both population and GDP are expected to be affected by
extreme floods. Affected population and GDP exposed to flooding would be higher with the
expansion of inundation area as a direct result of sea-level rise.
The number of affected population under RCP scenarios of 2.6, 4.5, and 8.5 is shown as Fig.
5a. Expected population magnitudes, which would suffer from 50 to 1,000-year CEWLs, range
between about 70,000 and 79,000. In both 2050 and 2100, this increment is sharp with an enlarged
recurrence period and the maximum increment of affected population approaches 20,000 in 2050
and 30,000 in 2100. Considering the intermediate scenario of RCP 4.5, around 5.57% to 12.36%



more people would be confronted with the inundation risk in 2050, while the affected population
would increase 9.52% to 23.53% in 2100. Detailed data of the increase in affected population are
provided in Table S1(c).

Similarly, sea-level rise also leads to an increased GDP exposure; the scope of affected GDP is

presented in Fig. 5b. In the case of no sea-level rise, the total GDP of Rongcheng City at risk from
extreme floods would be between $1.72 billion and $1.88 billion. As inundated area increasing due
to sea-level rise, the change in affected GDP is obvious. By 2100, projections for affected GDP
increase from $1.82 billion to $2.23 billion. At the most extreme, under the high degree RCP 8.5
scenario, affected GDP would increase by approximately 20% by the end of the century. Additional
information about increases in affected GDP is given in Table S1(d).

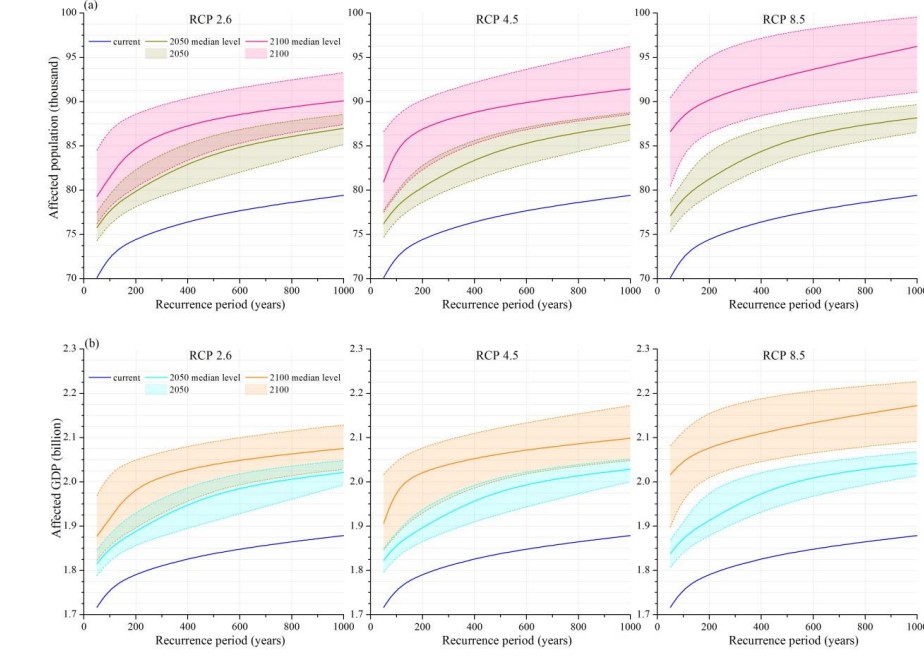



**Fig. 5** Affected population and GDP exposed to inundation in 2050 and 2100 under different RCP scenarios
**3.4 Variation of recurrence periods due to sea-level rise**
Refitting SEWLs combined CEWLs with future sea-level rise demonstrates that the recurrence
periods would decrease sharply due to climate change (Fig. 6). Results suggest that, by 2050, the
recurrence periods of extreme water levels would be shortened rapidly. For example, in 2050, the
100-year recurrence period for CEWL is likely to fall by eight years to 31-year (RCP 2.6), seven




years to 26-year (RCP 4.5), and five-year to 21-year (RCP 8.5). In 2100, more seriously, CEWLs
would be occurred more probably becoming common events under high degree RCP scenarios.
Among the different RCP scenarios, the shrink of recurrence periods under RCP 8.5 is more
significant than either RCP 2.6 or 4.5 scenarios. The worst case situation is that 1,000-year
recurrence period of CEWL would be occurred every three years; once in a hundred year events are
likely to become common, even occurring annually by the end of this century. Such recurrence
periods shortening would significantly increase the flooding risk over coming decades.

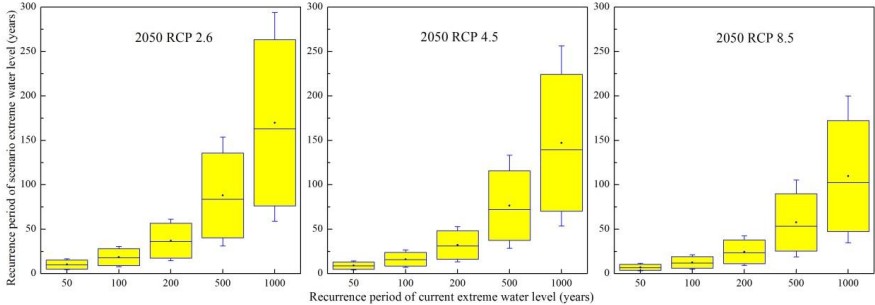


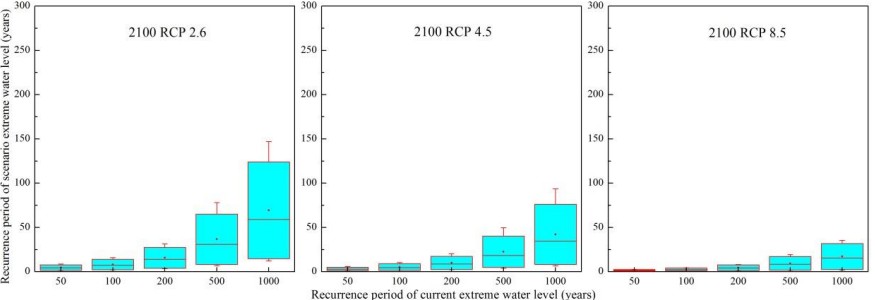


**Fig. 6** Variation in recurrence periods of CEWLs and SEWLs in 2050 and 2100 under RCP 2.6, 4.5, and 8.5
scenarios. In each RCP scenario, the variation in five representative recurrence periods of 50, 100, 200, 500 and
1000-year is shown. And the yellow boxes stand for the recurrence intervals in 2050 and the blue boxes stand for
the recurrence intervals in 2100. The data, presented the variation of recurrence periods, are just referred to
Chengshantou and Shidao stations.

**4 Discussion**
Based on previous studies of individual hazard and vulnerability (e.g. Li and Li, 2011; Wahl et al.,




2011), the extreme risk of inundation was assessed by integrating both of them. In this study, the
risk increase induced to sea-level rise was highlighted by the comparison of current with future
extreme water levels. SEWLs were recalculated by combining CEWLs with sea-level rise in 2050
and 2100 under RCP 2.6, 4.5, and 8.5. The results showed that recurrence periods would be likely
reduced by more than 70% by 2050 and this decrease could even exceed 80% by 2100 given high
RCP scenarios. In a similar study, Nicholls (2002) reported that a 0.2 m rise in sea-level could
markedly reduce recurrence periods of extreme water levels and a ten-year high water event was
converted into a six-month event. Indeed, as recurrence periods shortened, low-lying coastal areas
would have a higher probability of flood destruction over the next few decades.

The continuous sea-level rise would enhance the potential destructive force of future flooding.

For example, the results demonstrated that the potential inundated area would be extended by 3%
to 11% in 2050 and by 5% to 20% in 2100. In contrast, sea-level rise increased the inundated area
exposed to a cyclonic storm surge in Bangladesh by 15% with a 0.3 m rise (Karim and Mimura,
2008). Results showed that residential land and farmland were more vulnerable to sea-level rise
coupled with a large potential inundated area and a high proportion of expected direct damage.
Residential land was under the biggest risk, according to projected SEWLs under future RCP
scenarios which expected direct losses would up to $0.6 billion in 2050 and even exceed $1.00
billion by 2100. To put these predicted losses into context, average annual flood losses of Tianjin
City was estimated to be as high as $2.3 billion by 2050 (Hallegatte et al., 2013). It was predicted
that Shanghai, susceptible to high water levels, would be 46% underwater by 2100 with its seawalls
and levees submerged by rising sea-levels (Wang et al., 2012). A range of studies highlighted the
fact that many coastal cities, including San Francisco, would experience flooding in the near future
as a result of rising sea-level rather than heavy rainfall (Gaines, 2016). There was no doubt that
rising sea-levels would lead to a large number of people and property would be faced with flooding
risk, especially the fast growth of China's coastal cities (McGranahan et al., 2007; Smith, 2011).

291         Given the shortening of recurrence periods in future, property and assets exposed to extreme

floods would be more likely. For instance, results showed that under a RCP 8.5 scenario, an extreme
event that was possible to take place every 1,000 years and cause damage of $0.7 billion would
occur about once every 50 years by 2050, even once every two years by 2100. Under these
circumstances, many people and industries at extreme risk from floods would have no choice but to



retreat from coastal regions. However, studies indicated that most coastal populations were
completely unprepared for an increasing risk of extreme floods, especially in developing countries
(Woodruff et al., 2013).
Although this study manifested that sea-level rise would significantly increase the flooding risk,
some uncertainties still remain. First, on account of spatial heterogeneity, regional sea-level rise
should be projected in the future work. The objective of this paper is just to reveal the scientific
question that the impact of sea-level rise under global warming on extreme floods so that the
projection of global mean sea-level rise was used for its availability, which is consist with Wu et al.,
(2016). Nevertheless, there is no obvious land subsidence for the regional crustal stability. Second,
the combination of climate and weather extremes, including storm surges, astronomical tides,
rainfall and sea-level rise need to be focused on as they underlie and amplify the extreme events as
well as generating extreme conditions (Leonard et al., 2014). Because the coastal regions of China
have a monsoonal climate, combining inundation risk assessment with consideration of rainfall is
particularly important (Bart et al., 2015; Wahl et al., 2015). Third, human activities, which impact
on socio-economic development and alter feedbacks from climate change, are the mainly driving
force of future inundation risk (Stevens et al., 2015) and should be focused in the next research.
Consequently, the deeper exploration aiming at these uncertainties would be undertaken.

**5 Conclusions**

This study assessed the inundation risk resulting from extreme water levels with future projections
for 2050 and 2100 under different RCP scenarios. Results demonstrated that continuous sea-level
rise would augment the inundation risk by shortening recurrence periods and increasing the expected
losses and potential effect. (1) Sea-level rise would make low-lying coastal regions more possible
to be exposed to flood because of the recurrence periods shortening of extreme water levels. (2)
Inundation risk would be increased by the increment of inundated area, direct damage, and affected
population and GDP. (3) The analysis presented that sea-level rise principally threatened the vertical
land-use types for human survival, especially residential land and farmland. (4) Projections showed
that inundation risk would continue to increase up to 2100 and would be the most serious under the
RCP 8.5 scenario. In summary, these results revealed that sea-level rise dramatically increased the




flooding risk. Effective mitigation and adaptation plans are needed to deal with the increasing
coastal inundation risk.

## Acknowledgements

This research project was supported by the National Science and Technology Support Program of
China (Grant No. 2013BAK05B04), the National Natural Science Foundation of China (Grant No.
41301089) and the Clean Development Mechanism Funding Projects of China (Grant No. 2013034).
The authors also thanked Dr. Wenhui Kuang (IGSNRR, CAS) for providing the high-precision land-
use dataset of Rongcheng City.

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
