# Peer review of "Assessing the extreme risk of coastal inundation due to climate"

_Natural Hazards and Earth System Sciences, 2017_

## Short Comment (SC1) · 3 Feb 2017

Good paper for the cities like Rongchen suffered by the sea level rise especially under extreme risk.

---

## Referee Comment (RC1) · Anonymous Referee #1 · 9 Feb 2017

The paper only looks into change into the effects of extreme flood levels, and has a very simplistic approach to calculate the damages. It does not have sufficient content to be published. My advice is to focus the paper on either:

1.    the technical part of extrapolating 1/1000 flood levels out of a short series of water levels, and how SLR will affect these return periods (also, how exactly increase the storm surges by physical processes, water levels close to shore are higher, so wave upset is also higher, so no linear relation between SLR and flood levels).    E.g.    http://dx.doi.org/10.1016/j.coastaleng.2012.02.009, http://dx.doi.org/10.1016/j.coastaleng.2014.01.001

2.  the socio-economic assessment and also evaluate adaptive measures.  For this

water levels that occur more often than 1/50 years are also needed, land use change, and depreciation rates. E.g. doi:10.5194/nhess-14-1441-2014 , doi:10.1007/s10113-013-0420-z

Several additional remarks and suggstions are included in the pdf. Also English should be checked, too many mistakes in the text, sometimes blurring the exact point that is made

Please also note the supplement to this comment:
http://www.nat-hazards-earth-syst-sci-discuss.net/nhess-2017-31/nhess-2017-31-RC1-supplement.pdf

—————————————————————

---

## Author Comment (AC1) · 9 Feb 2017

Thanks for your attention and please contact me in case any questions arise.

---

## Referee Comment (RC2) · P. Schmidt-Thomé (Referee) · 14 Feb 2017

Review Assessing the extreme risk of coastal inundation due to climate 1 change: A case study of Rongcheng, China Overall statement. The text has interesting aspects but the work needs polishing and focusing before publishing. The scientific approach to combine increasing sea levels with floods, as well as population and socio-economic changes gives a good overview on the potential impacts on future floods. But the approach lacks a clear focus. Meanwhile the statistical approach in general seems ok, the mix of variables, and especially terms, blurs the result and the scientific quality. What is also missing is the potential that climate change adaptation measures contain. This text assumes that nothing in respect to coastal protection or other protective

/ adaptive measures would be undertaken. This as such is fair from a statistical approach, but then statements such as an "increased necessity for disaster mitigation" should be avoided. Meanwhile "disaster mitigation" is not necessarily only reactive, it has a certain connotation to this direction. Other proactive measures exist. These do not necessarily need to be assessed and discussed in great detail - that would go too far. But they should be mentioned. It should also be mentioned more clearly that the dramatic future effects of floods presented in this text are mainly encountered under the most extreme scenario, RCP 8.5. Theoretically the likelihood of this scenario is equal to the three other three main RCP's. So the statistical probability of the statements should be clearly mentioned, i.e. that there is a 75% chance that these effect could also not occur. Title (and throughout the text): What is the definition of "extreme risk"? Overall on the use of the term risk: The terms vulnerability and risk are not clearly defined. Vulnerability curves are mentioned, but it is not clear how these are used to define risk. What is the definition of the risk? The vulnerability and risk variables change throughout the text. One definition of risk is, e.g. Hazard x vulnerability = risk. How is the vulnerability defined here? What are the vulnerability variables? The flood prone area (extension), population, overall damages, farmland, or all together? Concrete comments Line 40 – 41 Why only disaster mitigation (=reactive)? What about climate change adaptation and, e.g. land use planning, coastal protection, etc (=proactive)? Line 42: Similar to risk above: Extreme water levels as such are no risk. These are hazards. The risk is based on vulnerability. Needs to be defined. Correct throughout text. Line 42 - 43. This statement is incorrect! There are plenty of studies and publications of flood prone areas changing under slr. See for example this book that contains several examples on climate change adaptation under changing sea level from several countries: Schmidt-Thomé, P. & Klein, J. (editors). 2013. Climate Change Adaptation in practice – From strategy development to implementation. Wiley Blackwell. Line 69. Here a short discussion on adaptation measures could be useful. Line 159. What kind of adaptation measures? Line 198 Here only flood damage and losses are mentioned, no risk. The terminology should be streamlined. Line 220, see comment above Line

244: Increase in frequencies of floods: Please add scientific references. Is the increase in occurrences only based on the fact of slr, or does it include and overall increase in frequencies? This issue is highly uncertain!

———————————————